

# Strengthening interdisciplinary water research – learnings from sports team management

Maija Taka[1], Katri Eeva[2], Maria Törnroos[3], Olli Varis[1]

5 [1]Water and Environmental Engineering Research Group, P.O. Box 15200, FI-00076, Aalto University, Finland
[2]Faculty of Education and Culture, P.O. Box 700, FI-33014 Tampere University, Finland
[3]Department of Management and Organisation, Hanken School of Economics, P.O. Box 479, FI-00101 Helsinki, Finland

*Correspondence to*: Maija Taka (Maija.taka@aalto.fi)

10 **Abstract.** Well-functioning teams with clear roles and advanced processes have a high potential to initiate peer learning and thus interdisciplinary collaboration. The need for interdisciplinary excellence is a modern-day phenomenon that characterizes all research, including water research. In this paper, we argue that by focusing on developing team culture and practices, a research group enhances their peer learning and psychological safety within and beyond the group. We approach this issue by summarizing the key findings from a five-year team development project in water research, where the data 15 collection focused on co-creation practices, active reflection, and journey mapping methods. These findings were described through a sports team framework and presented through Tuckman's team development model to capture the whole life cycle of a team. We present a collection of hands-on team practices that improved team performance and psychological safety by enhancing peer learning and utilizing the diverse competence of individuals. A diverse team with a hybrid hierarchy, transparent communication, and co-designed collaboration practices turned out to be important to strengthen commitment, 20 belongingness, and psychological safety. These were critical especially for doctoral students who were actively supported and encouraged for risk-taking and innovative, interdisciplinary research openings in water research. We conclude that coordinating research group activities that promote collaboration, diversity, and psychological safety can efficiently leverage interdisciplinary academic and educational performance.



# 1 Introduction

The surge of societal and environmental problems creates an increasing pressure for creative interdisciplinary collaboration in water research (United Nations, 2021). Compared to the mono-disciplinary approach, it is known to produce more innovative outputs, competent professionals to support efficient water resources management (Braimoh and Craswell, 2008; Carr et al., 2017), and the impact multiplies with the size of the group (Börner et al., 2010). The new technologies and modern research provide numerous opportunities to lever and expand inter-and transdisciplinary collaboration (c.f. Milojević, 2015), but much potential remains untapped. Scientific independence and difficulties in sharing tacit knowledge together with organizational barriers often challenge collaborative efforts (Bos et al., 2007). However, collaborative learning provides high social capital (e.g. knowledge, practices, and skills) both within and between the groups (Mäkelä, 2007).

The modern work life calls for new kinds of competencies (European Commission, 2022) and facilitates ways of working that may not be familiar in the academic communities, often being just nascent and finding their shape. Agile and deep learning are already cornerstones for professional success in knowledge-intensive jobs, but the learning to learn paradigm is widely setting aside the aged learning to do approach to education and professionalism (Bormann et al., 2017). This all culminates in education challenges: how to create a team that facilitates learning through diverse and engaging collaboration?

The circumstances that enable collaborative research processes are at the core of emerging fields of team science and group development (Bonebright, 2010; Börner et al., 2010). The idea of applying work teams beyond sports and military settings was first studied already in the 1950s (Sundström et al., 1990), but research on the benefits it has on interdisciplinary academic research teams is novel. The rising understanding about the importance of leadership in the industry gave rise to sport management studies in the 1970s, leading to an increasing focus on transactional and transformational leadership by the 1990s (Peachey et al., 2015). Objectives of sport management studies range from improved game strategies and decision-making (Kinnerk et al., 2018; Lord et al., 2020) to capacity building (Edwards, 2015) and psychological connection (Funk and James, 2001).

The sports team framework holds high potential for research groups and doctoral education development as well: it can improve and create new processes of peer learning and collaboration, and clarify the need for diversity, support, and shared motivation. Understanding group dynamics and the influence of a group for both individuals and communities is key for designing a well-functioning team (Forsyth, 2014). In this paper we utilize the sports team development framework in a research group context to address two research questions: 1) what are the key peer learning practices from sports management research that can strengthen team performance and excellence in interdisciplinary research in water; and 2) what are the best strategies to implement these into an academic setting? The findings are analyzed by using the world-famous group development model by Bruce W. Tuckman (1965). Despite being based on 1960s literature, it continues to be relevant for today's social settings (c.f. Natvig and Stark, 2016). This composite model conceptualizes the temporal changes in social and task realms as well as small group behavior, thus capturing the main efforts into five stages (Figure 1; Tuckman



and Jensen, 1977). The model of a linear process fits well to conceptualize a research process, such as a doctoral thesis journey from a research group perspective.

## 2.  Data and methods

### 2.1  Team management literature on sports teams

Journal articles on sports teams and their management were synthesized for creating a comprehensive understanding of the key modalities and practices that sports teams have. Scopus search was used for keywords on *'sport\* management\*'*, *team\** and with the rising themes, such '*belongingness*', 'sense of belonging', or 'stakeholder\*'. We excluded individual sports and

school and children sports from the analysis and focused on journal papers on professional sports teams. The majority of studies focused on football/soccer teams that well illustrate a diverse and competitive team with strong cohesion (Lord et al., 2020; Silva et al., 2020). The final synthesis is based on 42 journal articles.

### 2.2  Case study on water engineering

Peer learning and peer support practices were developed and studied in a five-year empirical research project on water

research (years 2017-2021) (Taka et al., 2021). The Majakka project ('lighthouse' in Finnish) focused on developing the culture and the practices of a research group in the context of doctoral education and research excellence in interdisciplinary water engineering at Aalto University in Finland. The core of the project was a team of six doctoral students with interdisciplinary research topics, their supervising teams, and a postdoc coordinating the project. The project provided a case study of an ideal setting with fully-funded students and a community that has for long focused on collective research success,

wellbeing, and equity. The culture-creation and practices were initially developed and piloted in this group of six doctoral students and then upscaled to Water and Development Research Group (WDRG) in a research group of ca 30 members, and into a larger Water and Environmental Engineering Research Group (WAT) of ca. 60 members.

Journey mapping workshops were organized for 1) Majakka doctoral students (N=6); 2) a control group of other doctoral students from the same research group (N=4); 3) doctoral students' advisors (mainly postdocs, N=6); and 4) professors

managing the research group and supervising the students (N=3).  Empirical data from Majakka doctoral students were continuously collected in e.g., objective discussions, reflective workshops, and co-designing workshops. In the results, all the direct quotes from the interviews and workshops are indicated in the text with quotation marks. The whole WDRG research group, where Majakka was included, was studied in regular workshops (twice per year) and with surveys on collaborative activities, group culture, and wellbeing. The data has been anonymized, thus individuals cannot be identified from the data.



## 2.3 Tuckman's group development model


Tuckman's five-phase model was used for analyzing the results. The *forming phase* is highlighted across the team development models, indicating a process of identifying the boundaries of interpersonal and task behavior and dependencies (Tuckman and Jensen, 1977). These link to two key factors that explain a team's success; an established identity (Thomas et al., 2017) and cohesion (Pescosolido and Saavedra, 2012). The social cohesion with communication, trust, and shared

understanding engages the team members to develop their processes and strategies together (Eccles and Tran, 2012) and enables agile and experience-based strategies (Pescosolido and Saavedra, 2012; Silva et al., 2020). However, it also holds a downside of creating informal group pressure and controlling the group members (Langfred, 1998).

The second phase, *storming*, is the first true test for the team, focusing on assessing interpersonal issues and the potential for conflicts. It also provides a critical process to tackle the potential resistance to group influence and thus opens the path for

the third phase, *norming*. Here the team focuses on the development of ingroup feeling and cohesiveness. New standards evolve and new roles are adopted. The *performing* phase concretizes the success of the preceding stages: interpersonal structures and group energy are the fuel for performance and activities, while they also support highly dynamic teams by allowing flexible and functional roles. The original four-level model of Tuckman was later fulfilled with the last phase of *adjourning* to complete the team journey towards wrapping up the work and then dissolving (Tuckman and Jensen, 1977).

This phase is highly relevant for sports teams with fixed seasons, and research teams with a project-oriented schedule.

## 3. Results and discussion

The sport management framework provided an extensive array of critical mechanisms, practices, and boundary conditions for a successful team. We identified critical themes that cover the main lateral (from individual to the community) and

temporal scales (from one routine to long-term strategies) of team management supporting the joint design, communication, implementation, and reflection towards the overarching goals of the team. Many of these mechanisms and practices are already utilized in academic communities, yet without acknowledging the wide potential of the rich documented experience on improving the culture of collaboration and psychological safety. The concepts of peer learning and shared goals in sports teams turned out to provide concrete approaches to focus on the interdisciplinary team instead of traditional single-

disciplinary characterized by cognitive boundaries that prevent knowledge sharing (Mäkelä 2007).

In the doctoral education context, traditional one-to-one supervision and the single-discipline community may hinder interdisciplinary research. The doctoral thesis topics per one supervisor may range from solar disinfection technologies to sustainable migration and from drought management to global food trade. As the research topics may go beyond supervisors' expertise, the main challenges the supervisee's experience are related to the project (e.g. scope, data, and methods) and self-

doubt (Barry et al., 2018).



### 3.1 Forming

***Cohesion vs. competition***

In the Majakka Project, the very early action towards team success was recruitment that focused on forming a group with a diversity of backgrounds, motivation, research topics, and competencies. The team was designed to represent a group of individuals with research topics that were diverse enough to reduce competition, but similar enough to ensure the need for collaboration. Diversity was present in various contexts: the research topics were diverse, students had backgrounds from water engineering to social sciences and geoinformatics. The diversity in work experience proved to be critical in co-learning transferable skills, such as time management and learning to learn.

Uhlenbrook and Jong's (2012) T-shaped competency in water research was used as a basis for designing the Majakka team for peer learning. Water, forming our core expertise, was researched in the project for example in the contexts of drought preparedness, climate oscillation, and food shocks; data-poor regions' sustainable water management, environmental migration, and urban floods. This introduced a diversity of theories, methods, research traditions, and novel research openings, resulting in increasingly interdisciplinary research in water. The group of six doctoral students and a postdoc collaborated with a diverse professional group: during the project they published 32 journal articles with 105 co-authors, who represented the whole globe and disciplines from social sciences to meteorology and from soil ecology to uncertainty modeling. On average, each paper had 6.4 co-authors. The diversity and thus its benefits are ever wider in the research group of 30 or 60 members. Notably, while helping doctoral students to define their own identity and position in the research group, it simultaneously challenged their identity building in the global scientific community. In the group of six doctoral students, this forming phase later strengthened intra-group interactions, peer learning, and cohesion.

In the beginning, aligning Tuckman's theory, the people were polite and getting to know each other: at first, the doctoral students experienced active grouping as artificial, but this quickly defined their group identity and home base for the whole thesis journey. The group defined their name, logo, and received peer support from both this small team and from the whole research group. The importance of this group became even more evident after the "honeymoon phase" of the thesis process.

***Diversity as a strength***

A season-long study on professional sports teams highlighted how these dynamic and diverse teams require frequent reinforcement of roles and togetherness, especially during challenges or changes in a team (Morgan et al., 2019). Our five-year study of the WDRG research group highlighted the highly dynamic nature: due to project-oriented work and funding the group doubled its size, while twelve researchers left the group and twenty new ones joined. Diversity in the group increased, became more valued, and favored new interdisciplinary research collaboration. This diversity, be it in football or research group, should be balanced with interpersonal similarity, holding a higher tendency for interactions, knowledge sharing, and clustering (Mäkelä, 2007). In practice, this balance was coordinated through skill mapping that helped to identify the gaps in



competencies. Those gaps were then filled by staff training and new recruitment. This reinforced the group's holistic motivation, shared goals, and the culture of life-long learning.

Our findings in interdisciplinary stakeholder collaboration for fostering diverse peer learning were controversial: for some research topics, the collaboration was easy to initiate, and the engagement was high throughout the project. On the other end, the practitioners were sometimes uncomfortable with the highly academic setting: despite clear roles and processes, the format of scientific writing or sharing their own data may scare the collaborator away. One solution for holistic benefits would be to initiate collaboration already in the recruitment phase and enable participation in the preparation phase.


### *Team management*

Team building in successful sports teams is coordinated by a skilled and transformational leader who concurrently focuses on individuals and the team, and invests in culture-building (Peachey et al., 2015; Fransen et al., 2020). The diversity and self-directed peer learning call for agile supervision that focuses on enabling and supporting practices. We observed that

during the forming phase, the supervisor's ability to understand the various starting points was critical. In sports, team building is based on acknowledging the individual's talent and potential (Sarmento et al., 2018), and their previous experience (Sinclair, 2010) to ensure that they are committed to working with people they do not know well yet. A similar need to test the team members' long-term committed was performed by the supervisor of the Majakka project. Supervisees with longer career histories had a better-defined need for the degree and competence, they understood their role in the team,

and they needed less support on, for example, collaboration and project management skills. Despite their different needs, equity among the team members was ensured in supervision.

Finally, the project piloted diverse supervision teams for doctoral students. The Majakka team was formed to represent diverse expertise in the field, methods, and practical applications. The plan was to provide continuous and highly competent support and peer learning, not only for the supervisee but also for the supervisor. In addition, students were encouraged to

invite new coauthors to every journal article based on their learning objectives. This led to situations of peer learning in terms of scientific practices, new methods, and transferable skills.

## 3.2 Storming

### *Interdependencies*

A research team could be illustrated as a football position map: showing a network of individuals with specified roles and

responsibilities that are critical in reducing interpersonal risks, competition, and contradictions while increasing togetherness and the selfless team culture (Morgan et al., 2019; Valentine and Edmondson, 2015). What it does not show is people's potential conflicting working styles and doubts about the team.

In our study, storming was included in the introduction process. The active attempts to value diversity were balanced with a critical assessment of potential conflicts. Long-term funding, flexibility on the research plan, and a strong emphasis on





individuals' learning and career goals set the basis for long-term engagements, however, the possibility for discontinuing their studies was present. To minimize risk for this, the supervisor team organized several mini-workshops during the orientation phase to define, value, and make use of the diversity (Figure 2). Potential for peer learning was identified and people were encouraged to update their research plans together to better utilize this high potential. Notably, group storming was more evident in the WDRG research group, and the non-believers or late adapters collaborated with lighter commitment

or not at all.

Interestingly, for some individuals the interdisciplinarity of the research group caused storming: early-career scientists with different backgrounds, especially from non-engineering fields, found it odd to work in an engineering research group. Especially for people joining the group, this "designed diversity" helped them to understand their unique role in increasing the diversity and introducing new approaches to the group. Communication was critical in the process that took a few

months.

### *The non-linear process*

The lack of formal support and unclear strategies are known to challenge doctoral students' learning (Bastalich, 2017). To complement this, we identified how critical it is to acknowledge the nature of a non-linear process already during the

storming phase. Research is characterized by high uncertainties that are often caused by external factors, not by the lack of training. *"Failure is just part of science. If you don't get rejections, you don't aim high enough"* one of our professors concluded. Unfortunately, we observed how experience and resilience usually go together: early-career researchers learn about this non-linearity and failures by experience, and thus may not be willing to take risks without sufficient resilience and knowledge.


### 3.3 Norming

### *Group culture and a long-term vision*

This phase was the most critical part of our empirical study, as the commonly agreed group culture encouraged its members for proactive participation and continuous collaboration for the common good, with the focus on enabling learning-centric

activities. Participation and co-creation were commonly agreed on norms and a collective agreement of investing 5% of work time for the common good was set up. This helped to overcome the assumption that collaboration creates extra work. Investing one or two hours per week in the group was paid back by support from everyone in the group. The activities favored collaboration over competition, leading to efficient sharing of ideas, contacts, and data.

Defining the culture set the basis for defining the team processes, that were then coordinated and implemented by roles.

Similarities to a sports team are numerous: high resilience is based on collectively agreed team vision and processes to implement it (Morgan et al., 2019). The sense of belonging and a shared sense of 'us' are critical for both personal growth





and engagement to the team (Thomas et al., 2017). This is well-known in doctoral education as well (O'Meara et al., 2017). In sports teams, psychological safety was found to simultaneously increase the team's inspiration, resilience, and performance, and to support individuals to speak up, provide input, and take risks. This resulted in an optimally functioning

and healthy team (Fransen et al., 2020). In Majakka, a key factor for psychological safety was the shared office and the group of students who started their degree at the same time.

### *Co-created strategies and procedures*

Sports management holds successful examples of teams with a culture of shared leadership: informal leaders (such as

captains) support the coach and provide great benefits through knowledge sharing, organizational creativity, and enhanced member experiences, among many others (Fransen et al., 2020; Kang and Svensson, 2019; Thomas et al., 2013). Such a shared and solution-oriented decision-making was introduced into a research group setting by using the subsidiarity principle: decision-making should be taken to the lowest appropriate level and closest to those they are affecting (UNDP, 1999). The leadership was defined as 'hybrid hierarchy', where the group has a low hierarchy as a whole, but when in need,

the leading professors provided supervision for everyone. The group's management was shared among three professors and assisted by a few postdocs who had responsibilities as employment superiors or facilitators. This enabled us to avoid endless loops of indecision while individuals were all equally respected and included. Based on the journey mapping analysis, this was a critical culture for creating psychological safety and encouraging risk-taking in research.

Research, including doctoral dissertation, requires long-term motivation, fueled by intrinsic motivation to learn, discover,

and develop. Even though interdisciplinary researchers may invest unsustainable amounts of energy in their collaborative work, they may still feel homeless in a strongly disciplinary community and lack formal recognition (Delamont et al., 2003; Wallen et al., 2019). We observed similarities with reported motivation factors of sports teams, that stems from a combination of inspired and motivated team members excited to take the journey together. These, together with a supervisor boosting motivation when needed, lead to increased resilience (Morgan et al., 2019). In the Majakka project, the culture of

"us" was critical for defining the culture of peer learning and belongingness.

Additionally, we discovered that doctoral students' long-term motivation is best nurtured by balancing scientific significance and stakeholder collaboration (extrinsic motivation) with students' objectives (intrinsic motivation). The elevated risks in interdisciplinary research were balanced by the team members who ensured that diverse support was always available. Students were encouraged and supported to invite external coauthors to their papers, and this networking was based on

active learning goal mapping. Doctoral students were encouraged to create new research openings and their attempts were respected. Despite the Majakka and the control group, doctoral students all had their high intrinsic motivation for research, these mutual agreements on for example co-learning and networking were critical in enabling continuous and diverse learning. These shared attempts characterize successful sports team as well (Kinnerk et al., 2018).



### *Map of roles and positions*

The mutually agreed roles outline the team's processes that strive towards efficacy, self-esteem, meaning, and continuity (Thomas et al., 2017). For an individual, the role encodes their responsibilities and interdependencies in the team, strengthening shared understanding of the collective accountability – even between strangers (Lusher et al., 2010; Ribeiro et al., 2017; Kinnerk et al., 2018). Professional sports players train for several positions to concurrently improve their performance and coordinate collaboration (Sarmento et al., 2018). We piloted this with doctoral students, who got responsibilities in thesis advising, teaching, and coordinating collaboration. This strengthened the feeling of belonging and encouraged individuals to take action for development. The collective responsibility for this culture strengthens team scaffolding, thus leading to actual processes for peer support and learning (Valentine and Edmondson, 2015).

## 3.4 Performing

### *Team actions for continuous development*

The collectively agreed norms towards active research collaboration and peer learning with equal investment set the basis for practices in the Majakka team. The established culture allowed its members to feel comfortable in taking action, sharing struggles and difficulties related to work and wellbeing. This strengthens the group cohesion that has become increasingly important – in both sport and research team settings – as the competition has been shifting from individual to institution-level (Musselin, 2018). The success becomes collective.

We rarely appreciate how one successful action, be it a goal or a published paper, is based on hundreds of hours of practicing and repetitions to develop technical, tactical, and physiological skills (Sarmento et al., 2018; Stojanović et al., 2018). Additionally, the star players are typically highly accurate and adjustable to complex situations (Bennett et al., 2019; Silva et al., 2020). These merited colleagues inspire, engage, and help the younger ones to get the ball.

The studied research group wanted to make use of this elevated cognitive capacity to decisions and actions, analogous to sports teams (del Campo et al., 2011). The research group aimed to make this nonlinear process more visible by facilitating activities where the more advanced researchers provided experience-based support, tips, and guidance for young scientists. Improved and more systematic internal communication and for example, weekly group seminars set the basis for this communication. We piloted processes where the advanced researchers were active in initiating the collaboration, giving gradually more responsibility for the younger ones. Postdoctoral researchers were often appointed as doctoral thesis advisors, which provided them with a well-recognized need to learn about research, leadership, and pedagogics. One doctoral student observed how these activities for psychological safety created a moment when they *"stopped worrying and started enjoying"*.

The process of continuous learning was important to communicate, as doctoral students and non-academic partners may assume they need to master skills, such as scientific writing *a priori.* This had a high potential in hindering or even canceling



collaboration. Additionally, the importance of informal meetings was observed and became even more evident when the global pandemic forced people to work remotely. The peer learning was facilitated in alumni meetings, where 'alumni of the month' shed light on their current work and their doctoral thesis experiences, bridging the degree into practical work.
Another form of cost-efficient collaboration – a low investment with potentially high gains - was the annual *'doctoral students meeting practitioners'* event, which provided one-time sparring discussion and networking, or even a beginning for a mentoring relationship between doctoral students and stakeholders. The session was a side event at an international conference.

Collaboration and peer learning helped to understand the norms and principles for behavior, i.e. *'situational probabilities*
*knowledge'*, meaning preparedness to the actions of a player to a given situation (Eccles and Tran, 2012). As doctoral students' key issues are related to mastering research practices and methods challenge them the most (Barry et al., 2018), and the merited supervisors have deep knowledge of students' typical reactions to these (for example how a student is devastated by a major revision from a journal), we wanted to provide this knowledge for the whole team. Research is about taking risks and researchers are encouraged to go beyond their comfort zone, test, and expand their abilities, and seek endorphin peaks in
their work.

In the Majakka group of doctoral students, the strong cohesion initiated several types of collaboration. Students identified mutual needs for learning new research methods and did analyses jointly for their research papers. The group published a journal article where they jointly learned about sustainability in water, learned about sustainability in water and new quantitative analysis; learned how to conduct a systematic literature review; and gained knowledge about the publication
process in general. The students peer-reviewed each other's papers, wrote joint papers, and attended conferences as a group. Teaching was often done together. They represented a team that was active in networking, stakeholder engagement, and external communication. Presenting as a group helped to initiate new activities.

*Well-planned strategies and procedures*
The members of the studied research group agreed that their collective strategy was to focus on advancing interdisciplinary research collaboration and to build up their international excellence with an impact beyond academia. Group members wanted processes that enhance life-long peer learning, strengthen communication (both internal and external), and foster the culture of transferring knowledge and networks. The data from Majakka doctoral students illustrated how individuals' learning leaps, failures, and ability to take responsibilities vary in space and time (Figure 3). However, we identified
processes that were universal to all students, irrespective of their research topic. Communicating such generic processes to new doctoral students and thesis advisors increased understanding and risk-taking while decreasing the risk of misunderstandings. As failures and drawbacks are an unavoidable part of both sports and research, we should focus on creating scaffolding and resilience to encounter them. Both long-term vision and interdisciplinary collaboration increased support for the individuals.




### *Puuhas for collaboration*

Participation in several journal papers with varying roles helped to understand the roles and pressures of others and enabled interdisciplinary research collaboration. Journey mapping results highlight how doctoral students enjoyed the most those papers where they were working in an interdisciplinary team. At the beginning of their studies, the position of not being the

corresponding author was found to increase their belongingness and psychological safety. It supported the phase of defining their researcher identity, deepening their knowledge in the field, and learning about the publication process. The psychological safety enabled new research openings and the group for example decided to collect a novel global dataset on migration – the risk for this was lower as the group of doctoral students were working together. Regarding the growth as a researcher, ownership of their work, and an established role in the community, the 3-to-12-month research visits were found

valuable. These external experiences are known to be critical for creating an expatriate experience and the ability to take different perspectives (Mäkelä, 2007), which is highly relevant for fostering interdisciplinary research.

Similar to a sports team (Bruner et al., 2014), we designed an ensemble of harmonized processes aiming to concurrently develop skills and advance psychological safety, and build resilience against drawbacks. We piloted the processes for interdisciplinary peer learning and support through two lenses: first, we designed processes based on the stage of doctoral

students. '*Synthesis group*' was a small group for doctoral students across the research group finishing their degree, writing the synthesis of their article-based dissertation, and actively finding the next step in their career. This group turned out to be a critical source for support and knowledge, and for tackling *"the existential crisis about the dissertation"*, as some students expressed. During the project, we facilitated five generations of synthesis groups and created an online collection of resources of e.g., pre-examination comments, documents, and practical tips. Each group contained three to six doctoral

students who met regularly to communicate their work and set the next goals – the more multidisciplinary the thesis topics, the more successful the group. A postdoc coordinated the meetings and knowledge co-creation. A similar small community, *Rookies club',* was piloted to support the youngest researchers to initiate their academic careers. The group focused on basic research training, peer support, identity building, and knowledge transfer. Both the *Synthesis group* and the *Rookies club* have been highly successful informal communities in strengthening resilience and further elaborating the goals of the

doctoral degree. They also point out that success is always made in teams - whether in sports or research.

The second lens for piloting the collaboration processes was designed for the entire research group including everyone from research assistants to professors. *'Puuhas'* ('puuha' is a Finnish word for a light everyday task; in plural, 'puuhas') structured the teamwork. The group collectively identified the key processes common for all research and thus established jointly facilitated activities on external communication, skill clinics, and weekly research seminars with both internal and external

speakers. Each *puuha* was coordinated by a group of team members who were responsible for organizing activities during their term. This aligns well with Tuckman's model where the performing phase provides an opportunity for the coach to delegate more work for the team while focusing more on developing the people and the group. As responsibilities are shared, the conflicts are easily solved without the coach's intervention.



Initially, people sought for balance between their research and *puuhas* but taking a risk for piloting brought these two closer

together in just a few weeks. For doctoral students, *puuhas* provided support to develop their general work processes which were increasingly important once the 'thesis honeymoon phase' was passed. We successfully piloted upscaling of the *puuha* co-creation culture to two other research groups, and the efficacy of such an upscaling phenomenon was one of our most tangible positive surprises within the entire project. This firmly encourages us to continue upscaling with this method to the school and even university levels.

It was evident that *puuhas* required careful planning and reflection: the number of activities need to be limited; the likelihood of failure was high if they needed too much investment or lacked participants. Routines and regular activities appeared critical. Those *puuhas* that turned out to be unnecessary or contradicting were terminated. One example was the Monday meeting which aimed to increase intragroup communication to identify ad-hoc needs for peer support. We soon identified that these meetings were sensitive to the size of the group: a group of fifteen members created unnecessary pressure instead

of psychological safety that was the initial goal. We decided to continue the meetings in smaller groups and concentrated the help requests to the teams' online platform.

### *Reflection and video analysis*

Following the sports team framework and their practices for active reflection and dialogue-based development (Pescosolido

and Saavedra, 2012; Silva et al., 2020), we wanted to foster reflection by co-designing mechanisms to benefit both individuals and the team. The piloted reflection tools, such as journey mapping, turned out to reduce the competition among individuals and helped them to understand their own, continuous development, especially when the development was other than publishing a paper. Defining personal goals and milestones turned out to be challenging, and support from more merited colleagues was instrumental. However, visual mapping was found useful not only for communicating feedback but also for

reflecting own viewpoints on the process. Illustrating the process helped to understand the need for interdisciplinary collaboration and pointed out the moments when the collaboration should have already been initiated.

Especially the successes and failures in the interdisciplinary collaboration efforts were valuable for finding fruitful ways of working and communicating. For instance, getting an interdisciplinary research paper accepted to a high-quality water journal is not easy and often leads to rejection. A young researcher may encounter this as a personal failure, but there

remains always much to learn from the experience. At group level, the team's tactical approach and *puuhas* were regularly, at least twice per year, reflected and iterated. As an example, in the year 2017, we had 19 *puuhas*, and in 2021 they are narrowed and merged into four. The process has become an inevitable part of the culture and operates with almost no extra effort. This process of reflection helps the team to build their mutual trust and team-regulatory, increases individuals' accountability, improves internal communication, and helps to iterate the processes (Morgan et al., 2019).






### 3.5 Adjourning

An Academic team is in continuous change as projects start and finish, and students graduate, and new ones enroll. Welcomes and adjourns are frequent. Sports teams' adjourning peaks at the end of their season before starting the forming phase to begin the next training and game season. In our research, the adjourning process provided challenges. While the doctoral education process should focus on future-oriented and proactive learning on research and transferable skills, such as teamwork. Assessing the success in these befalls during adjourning: students invest more in defining their professional identity and job hunting. Who are they, how are they received by a diverse mass of employers? For our target doctoral students, this was a wedge issue. They described the process of writing the thesis synthesis as either a moment of *"seeing light at the end of the tunnel"* or a moment of *"existential crisis about the dissertation"*.

Unfortunately, we did not use the full potential of adjourning in the interdisciplinary and peer learning setting. The interdisciplinary team did help them to diverse their understanding of different work possibilities, thus helping their career planning and competence building. Several alumni participated by providing insights from their careers. During the adjourning phase, the doctoral students were already very independent and proactive in contacting future employers and applying for postdoc funding. As they were also successful in those activities, most of the doctoral students were only part-time employed by the university during their adjourning phase. Growing from a doctoral student into alumni was found important and many of the research group alumni expressed that they understood the value of good supervision once they started in their supervisor position. Thus, having co-designed and well-tested processes sets a strong basis for initiating and maintaining diverse collaboration beyond the own research group.

### 3.6 Interdisciplinary collaboration – a hat trick?

The studied research group and the group of six doctoral students provide several examples of the importance of the scaffolding research group culture and thus positive impacts of interdisciplinary peer learning and collaboration. The sports team framework contributed with practices and resources that are critical in enabling successful and learning-centric teams. However, Tuckman's model has several weaknesses. The most important observation is the absence of peer learning, which was critical in sports teams and research groups. The linear model does not capture the oscillation that the research group dynamics actually holds. Further, the concepts 'team' and 'group' hold different meanings despite they are considered similar in this work (Fisher et al., 1997).

The analysis of sports team literature and empirical research on research group setting provides high potential in practices for peer learning, and examples of the threats and benefits it has on interdisciplinary research, learning, and psychological safety in the group.



## 4. Conclusions

Due to the increasing seriousness and complexity of today's water-related challenges, the societal demands for the highest level water experts are evolving fast towards enhanced interdisciplinarity. Doctoral education in water research should spearhead this development. A huge development need is obvious in terms of approaches, modalities, and practices for
interdisciplinary teaching and learning.

Sport team literature was used as a framework for identifying and co-creating practices into a research group. The five-year research of the research group provided new knowledge about the practices for enhancing self-directed peer learning, with a synthesis of the threats and positive effects on individuals, teams, and research.

We conclude that peer learning would benefit from mechanisms and practices that establish, reinforce and enable both
formal and informal activities. These would facilitate collaboration, new research openings, and strengthen psychological safety among the team. The established culture is easily adapted by new team members and the main storming occurs when the culture is initially created.

## Author contribution

MT and OV designed the research and data collection, and MT was responsible for the data collection and associated data analysis. All authors contributed to the study methodology development and manuscript writing.

## Competing interests

The authors declare that they have no conflict of interest


## Acknowledgments

Research funding was provided by Maa- ja vesitekniikan tuki ry. We thank the doctoral students in the Majakka project for their active participation, fruitful inspiration, and inspiring collaboration. The Water and Development Research Group (Aalto University) we thank for their innovative ideas and endless motivation for co-creation.




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





| Tuckman's original description | Orientation and testing, identifying the boundaries of interpersonal and task behaviors, and the dependencies among the group members | Identifying the conflicts and polarization among the individuals, resistance to group influence | Development of ingroup feeling and cohesiveness, new standards evolve, and new roles are adopted. Individuals express intimate and personal opinions | Interpersonal structure and group energy are the tools for performance and activities. Roles become flexible and functional. | Termination of roles, completion of tasks and reduction of dependency |
|---|---|---|---|---|---|
| | **FORMING** | **STORMING** | **NORMING** | **PERFORMING** | **ADJOURNING** |
| Rickards and Moger, 2000 | Orientation phase | Exposure and addressing personal conflicts | Establishing norms of behavior to subside the storm | Team efforts are directed towards tasks | Termination through task completion or membership disruption |
| Largent, 2016 | Leadership: explaining or directing. Participants with high enthusiasm and low skills | Leadership: demonstrating or facilitating. Participants with low enthusiasm and low skills | Leadership: guiding or coaching. Participants with rising enthusiasm and growing skills | Leadership: enabling or consulting. Participants with high enthusiasm and high skills | |
| Erenli and Ortner, 2011 | Especially testing and orientation were crucial stage for the highly motivated group | Distinct forming stage made storming insignificant, but it helped to observe emotional response reactions of the team members | The key stage: new standards were evolved and new roles adopted. Risk of a member losing their motivation due to an inability to gain "project speed" was highly evident | Easy to achieve, as everyone had sufficient knowledge to fulfill their tasks | |
| Bird et al., 2011 (illustrated as a linear process) | Selecting team members, depend on leader for decisions, low involvement, unclear objectives and direction, weaknesses are hidden | Concern for others, leadership challenged, risks taken, experimentation, support needed, feelings explode | Agree procedures and processes, ground rules established, conflicts discussed, wider options considered, compromise to achieve goals, more listening | Effectiveness is seen as a priority, trust within the team members, needs of all considered, maximum use of energy and talent | Labelled as "transforming": Re-establish goals and priorities, flexibility on process and leadership, flexibility of roles |
| Murray and Blackman, 2006 (illustrated as a circle loop) | Discovering acceptable behaviour and boundaries of task, developing team behaviours and focusing on the objectives | Hostility & conflict expressing individuality; task resistance. Opinions polarize | Group becomes cohesive and norms established and it exchanges ideas openly, the task ideas are harmonized and conflicts avoided | Social entity that supports activities with a cohesive focus, problem resolution with insight into task, problems and what is abnormal behaviour | |
| Cullen and Calitz, 2015 (illustrated as an elevating linear curve) | Confusion, uncertainty, assessing situation, testing ground rules, feeling out others, defining goals, getting acquainted, establishing rules | Disagreeing over priorities, struggle for leadership, tension, hostility, clique formation | Consensus, leadership accepted, trust established, standards set, new stable roles, co-operation | Successful performance, flexible task roles, openness, helpfulness, delusion, disillusion, and acceptance | Disengagement, anxiety about separation and ending, positive feeling towards leader, sadness, self-evaluation |

*Team effectiveness, or personal relations between group members, or self-reliance*

*Time, or task functions, or performance impact, or competence, or gradual release of responsibility*

**Figure 1: A visual synthesis of the interpretations of Tuckman's model (Bird et al., 2011; Cullen and Calitz, 2015; Erenli and Ortner, 2011; Largent, 2016; Murray and Blackman, 2006; Rickards and Moger, 2000).**





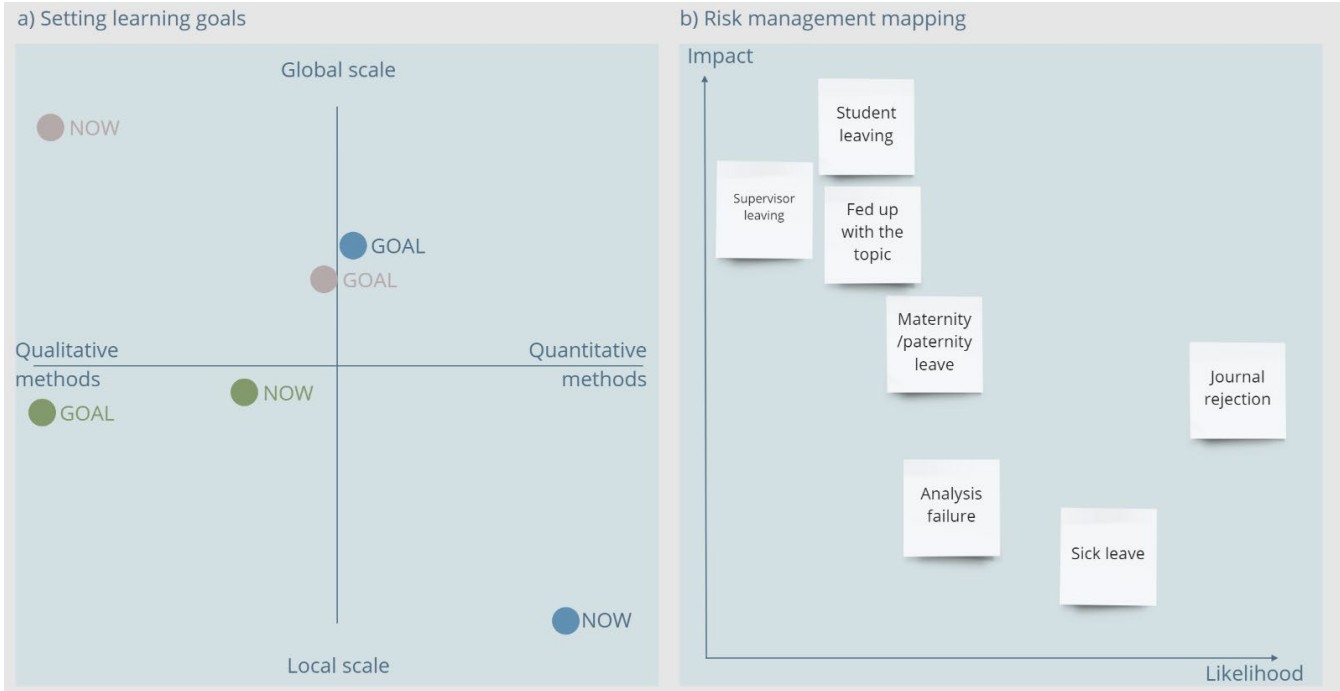

**Figure 2: Examples of the workshop tools for a) setting goals and path towards them, and b) to assess risks on the doctoral thesis process. Everyone positioned themselves into the map in (a) in the forming phase, and the output helped to identify the highest potential for peer learning and support. Mapping in (b) helped the team in storming phase: to acknowledge the risks, their likelihood, and to increase joint preparedness for those.**





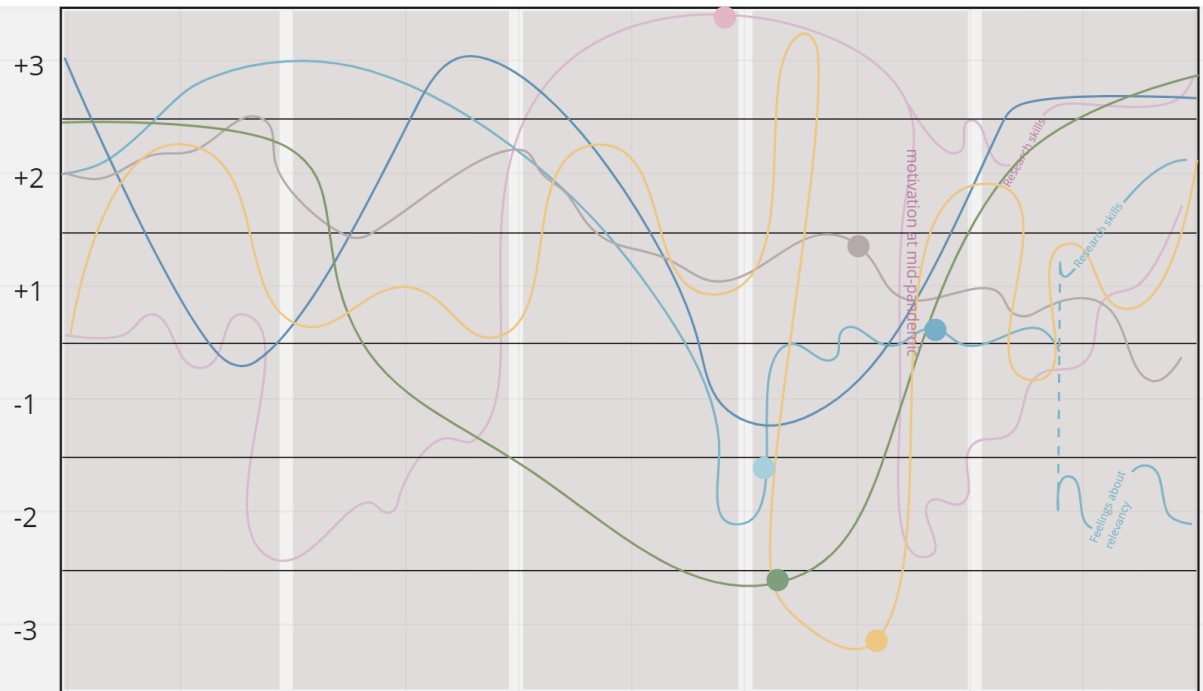

**Figure 3: Visualization of the six Majakka doctoral students' journey (N=6). The x axis represents the five-year journey and y axis the reflection of the emotions: students pointed out key activities, actions, and milestones and graded their effects following Nilsson et al.'s (2016) classification: the effects were either positive, i.e. enabling (+1), reinforcing (+2), or indivisible (+3) effect on the work and wellbeing. The negative effects were either constraining (-1), counteracting (-2), or cancelling (-3). Each line represents one doctoral student, and the circles represent the moment when the COVID19 pandemic started.**
