# Peer review of "Strengthening interdisciplinary water research – learnings from sports team management"

_Hydrology and Earth System Sciences, 2022_

## Author Comment (AC1)

**Aalto University
School of Engineering**

**HESS-2022-85**

**Strengthening interdisciplinary water research – learnings from sports team management**

We sincerely thank the editor, associate professor Hermans and the two reviewers for their careful reading and constructive comments on this manuscript. The reviewers' input is greatly valued and supports us to improve the work.

We will make major revisions to the manuscript, including:
1. Overall rewriting of the work, including descriptions of the project, used methods, and a more critical assessment of the sports team framework,
2. Reporting the sports literature review and the method description in a more systematic way,
3. Restructuring the Results and Discussion to align Tuckman's model more strongly.

These and other revisions are described in detail below in our point-by-point replies to the comments.

**Responses to RC1** (Erik Mostert, 'Comment on hess-2022-85')

1.0 The paper addresses the challenge how to create interdisciplinary research teams that facilitate collaboration and learning. It draws on the literature on sport teams management and discusses the experiences in the Majakka project, a five-year water research project in which peer learning and peer support were central. It is not a novel idea to compare teams outside of sport with sport teams, but if there is a substantial body of literature on sport teams, it is definitely a good idea to take stock of this literature, assess the relevance for interdisciplinary water research, try and apply the insights gained, and discuss the experiences. That being said, there are some major issues with the paper.

Response: We thank you for the critical assessment and for raising the major issues regarding the manuscript.

1.1 First, the paper does not **systematically analyse the literature on sport teams**. As a reader, I would like to know what type of research has been conducted on sport teams management and what the main conclusions or lessons are. Throughout the paper there are references to this literature, and at several places the authors state that a specific issue is equally relevant for sports and for research, but that is not quite the same thing. Many **terms and concepts** are used in the paper that are not properly discussed, such as social capital, transactional and transformational leadership and of course interdisciplinary. A term that is used several times later in the paper is **"psychological safety"**, but there is no discussion of this term and why it is relevant for interdisciplinary water research.

Response: Thank you for pointing this out. We will bring a more systematic literature review approach to the manuscript, providing a description of the review and a more holistic synthesis of the main findings. Further, we will pay special attention to describing the key concepts to the reader and focus on the use of the word "interdisciplinary". Finally, we will better argument our selected concepts and their use in this paper. (raised also in RC2's comment 2.19).

1.2 Secondly, the description of the case is rather abstract. For example, the paper mentions that students were encouraged to invite co-authors for their articles, which led to "situations of peer learning in terms of scientific practices, new methods and transferable skills." (line169-171) A concrete example specifying the practices and methods learnt and the skills gained would be really useful. In addition, more details about the Majakka project would be welcome. What was it about, was it one big project with each PhD student working on one part of it or was it rather a cluster of related projects, and was there only one supervisor for all PhD students? The teams building activities could be described more systematically and clearly too. Moreover, there is talk of the research group, but oftentimes it is not clear whether this refers to the Water and Development Research Group or the larger Water and Environmental Engineering Research Group.

Response: We will provide more concrete examples of these activities and cases. The research project designed several activities that are easy to initiate and that can provide significant support with very little input.

Additionally, we will provide more empirical cases of individual doctoral students, their systematic practices, and the resulting learnings. For example, one doctoral student coordinated the production of a global data set, and in this process, they developed their skills in risk and project management, internal communication, collaboration coordination, and data set production. Further, the inclusion of external coauthors provided wider support for the student and supported their deep learning on the topic.

To help the reader to understand the participants and roles, we will provide a table of the key participants in the revised manuscript.

1.3 Thirdly, the methodology is not explained well. In section 2.2 there is talk of journey mapping workshops, objective discussions, reflective workshops, co-designing workshop, regular workshops, interviews and surveys, but details are missing. What is the differences between the different types of workshops, were they recorded, transcribed or summarised, how and by whom? Who were interviewed how often and by whom, and what questions were asked or what issues were addressed? And how, and how often, were collaborative activities, group culture and wellbeing assessed in the surveys? Measuring particularly group culture is not a straightforward process and depends on one's concept of group culture. An further bit of information that is missing is the role of the authors in the Majakka group or the lager WDRG or WAT groups. If they were involved, this is something to reflect upon as it may influence interpretation.

Response: This is an excellent point – we will provide a summary table of the key methods with detailed information about the different data sources, data production, and analysis.

1.4 Fourthly, I can imagine there are also differences between sports and research, but these are not discussed. In team sports, teams always win as a team, and if a team members want to show off his prowess at the expense of the team, this is visible for all spectators. In PhD research projects, however, PhD students obtain an individual degree and collaboration takes place behind the scenes – or not. According to the authors, the Majakka project took place in a community that focuses on collective research successes. This not the same everywhere in academia. Discussion of the differences between sports and research and between different research settings could results in a clearer view on the challenges of interdisciplinaritry and the best strategies to promote it.

Response: This is true – we will provide a more critical analysis on this with a discussion on the differences between the concepts of "team" and "group". Additionally, Majakka was a part of a research group with an exceptional atmosphere and culture and might be biased compared to academic communities in general. We aim to present this more clearly and demonstrate how the key findings of Majakka could be applied to any research group.

1.5 A smaller but not unimportant issue is that Figure 1 is not clear. It is not clear what the axes represent and why references are given both in the caption and in the figure itself. The descriptions in the figure partly describe Tuckman's model or further developments of this model, and partly they seem to describe the Majakka case, or at least they are formulated as such (e.g. testing and orientation "were" a crucial stage).

Response: Figure 1 aims to provide a summary of how Tuckman's model and its phases have been described and illustrated in previous studies, as the definitions are diverse. References are both in the figure and in the figure caption, as the latter will provide direct linkages to the reference list in the online paper.

We will revise the figure and focus on clarity and readability. The content will be more synthesizing and less describing.

1.6 The conclusion of the paper is that "peer learning" would benefit from "mechanisms and practices" that establish, reinforce and enable "formal and informal activities" that "facilitate collaboration, new research openings and strengthen psychological safety among the team." (lines 414-416). This conclusion lacks specificity and it is not clearly supported by the preceding analysis.

Response: Aligning the earlier comments, also this concern will be covered by providing more concrete examples of the activities: what was piloted, by whom, how, and what was the concrete benefit. These practices include, for example, joint workshop, a collaborative journal paper, jointly advised master theses, online writing sessions, and ad-hoc skill clinics. Moreover, we will further discuss peer learning in this context and draw on the activities that illustrate the ways peer learning emerges in Majakka. See also our response to comment 1.2.

1.7 My own conclusion is that the paper needs to be rewritten completely. It is definitely worthwhile to analyse the literature on sport teams management and discuss the experiences with applying the lessons, but the result should be a quite different paper.

Response: Thank you for pointing this out. We are confident that by meeting all the above-mentioned issues the quality of the paper will improve significantly.

**Responses to RC2** ('Comment on hess-2022-85')

2.1. This manuscript uses a framework developed for examining the development of small groups, that has been further developed in the literature to explain the development of sports teams, to explore the development of collaborative PhD research groups. This is an interesting application of an existing framework that offers considerable potential to better understand how, why and when successful collaborative research can be achieved. The data set is centred on case study analysis of an interdisciplinary doctoral education pilot project (the Majakka project) using the sports teams framework, data from "journey mapping" by members of the doctoral project and the wider programme in which it is embedded, as well as interviews and workshops with the participants during the project.

The potential strength of this paper is the unique data set provided by the case study that has considerable value for improving understanding of how small collaborative research groups develop, can be supported, and what they can achieve in terms of interdisciplinary research and education outcomes. However, additional explanations and clarifications, some further data analysis and paper restructuring are needed to maximise the explanative potential of the data and its findings.

Response: Thank you for the thorough review work. The revision will consider all the issues raised in your review and in this letter, we provide a point-by-point description of how the work will be completed.

2.2. Firstly, in the introduction and throughout the paper, all key terms need to be clearly defined and explained: inter and transdisciplinary, collaborative learning, self-directed peer learning, social capital, transformational leadership, culture-building, group storming, active learning goal mapping etc.

Response: Thank you for pointing this out, Reviewer #1 raised this same issue (see comment 1.1). We will pay attention to describing the key concepts and their differences (for example collaborative vs. peer learning) and ensure that the reader has sufficient information on how the concepts relate to the research presented in the manuscript.

2.3. Some more robust arguments as to why the sports team framework was chosen would be beneficial i.e. did this come from the authors past experiences? The background to this needs quite some reworking e.g. paragraph 3 of the introduction: "The sports team framework holds high potential for research groups and doctoral education development as well: it can improve and create new processes of peer learning and collaboration, and clarify the need for diversity, support, and shared motivation". The framework itself cannot improve and create new processes etc. but it can be used to analyse how these emerge. The research questions also require some refinements as in the manuscript in its current form they are not convincingly addressed.

Response: We will rewrite the introduction to clarify the rationale and motivation for using the sports team framework here. We see that applying it to a research environment holds a high potential with various similarities to research groups; a diverse group of highly competent individuals, following a program-like training and activities, and how are highly skillful in collaborating towards a common goal. This would introduce various excellent practices and culture into an academic setting where individuals traditionally work very independently and are easily located in their own single-discipline silo. Finally, we will provide examples of how sports team culture has been applied in other fields.

2.4. Figure 1 should be introduced and much more fully explained, either in the introduction or in a new section 2, that specifically reviews the literature, develops the framework and places it in its new context of collaborative (and interdisciplinary) research. (Also, the references related to Figure 1 are missing from the reference list).

Response: This same issue was raised by Reviewer 1 (see our response to their comment 1.5). Figure 1 will be reproduced to better synthesize how Tuckman's model has been communicated and visualized in previous literature. The figure aims to provide the background understanding of the model and thus help the reader to understand what the different phases focus on and how they are causally linked. Additionally, we will pay careful attention on the references.

2.5. In the methods section, a more thorough description of the journey mapping workshops (who was there, what did they do, and when) and more specific details of the workshops, interviews and surveys should be provided along with how the data were collected (recorded, notes taken, photo of images and figures created) and managed (transcribed, categorised, sorted etc.).

Response: Thank you for pointing this out. We will focus on the method description by for example providing a summary table of the key methods with detailed information about the different data sources, data production, and analysis (see also comment 1.3 by Reviewer 1).

2.6. The results and discussion do not seem particularly well aligned to the Framework that has been proposed in Fig 1. Paragraph 1 seems unnecessary and could be removed.

Response: We will revise the text to ensure that the framework is used adequately. The subheadings/themes in each phase are defined by ourselves based on the findings of the empirical research. We will further clarify this in the manuscript.

2.7. It would then make most sense, if the sub-headings of each element of the framework were aligned with the key elements of each section.

Response: Will be edited as suggested.

2.8. Under 3.1 Forming, the subheadings of i) Boundaries, ii) Dependencies, and iii) Leadership, would be much more logical as these words appear in Tuckmans original description. Some more detailed analysis of research boundaries (how they are set up, what constrains them and how they are defended in an interdisciplinary setting would be extremely interesting). Under Dependencies, you can talk about how roles are allocated and human resources selected to build a "team". Here I would expect some discussion of how different disciplinary skill sets are brought together. Under Leadership you can bring together all the material scattered throughout the analysis and discussion on leadership, how it was structured and how it worked.

Response: As mentioned in our reply to comment 2.6., these subheadings were defined by our research findings and are concepts that we found from both sports teams' literature and in our empirical research. The work uses Tuckman's model and its phases but advances this by presenting the results through these "common nominators" that Tuckman did not use. We will pay special attention to renaming these.

2.9. Under 3.2 Storming, the sub-headings of e.g. Conflicts/Polarisation, and Facilitation. Figure 2 needs much more thorough explanation here. What do the colours relate to in Fig. 2a and why do the axes cover scale and qualitative/quantitative? In 2b) what data is this based on? Could you add a quantitative element, i.e. what was the risk that the greatest number of students were concerned about? Why is maternity or paternity leave perceived as a risk? This needs to explained and contextualised (for the Finnish setting).

Response: Good points – we will focus on improving the figure caption to help the reader understand the figures. These are presented to provide easy and hands-on tools for communication and planning both individuals' and the team's work. In 2b the sticky notes are examples from one workshop but no systematic research with for example control groups was done. These figures are not showing results, but instead, provide useful tools to support early-stage doctoral students.

2.10. For 3.3 Norming, sub-headings of Cohesiveness, New standards, and New roles, would shape a very interesting discussion. The term "culture" seems vague and does not fit with the sports management framework. More details on the mechanisms and how they were operationalised to achieve the "Cohesiveness" would be needed. E.g. how was a commitment of 5% of work time for the common good implemented and monitored? How was psychological safety achieved and what is the evidence of this? How was learning across different disciplines achieved? i.e. through joint fieldwork, shared data collection, different analysis approaches of the same data sets etc. What new standards of evaluation have emerged? How have new roles within the research groups evolved or been developed?

Response: Culture development has been studied in sports team literature, whether as a process of top-down development or co-development. We will provide more insights based on the analysis of annual discussions and interviews with the participants and

alumni. For the research group in question, this culture creation has developed a culture where individuals and the team create new roles based on individuals' competencies, learning goals, and the group's needs. All this follows the subsidiarity principle. We will pay special attention to reporting this all in a scientific manner in the manuscript.

2.11. Under 3.4 Performing, sub-headings of e.g. Interpersonal structure, Performance, Activity. Under interpersonal structure, you have great material about the interactions between more advanced researchers and early career scientists. More details and explanations of the findings of the interactions between academic and non-academic partners would be very valuable especially in terms of the challenges to collaboration that these may present. Figure 3 needs considerably more explanation. Explain the scales – what exactly is meant by enabling, reinforcing, indivisible, constraining, counteracting, cancelling?

Response: This collaboration with non-academic partners was truly a learning process for us – with some partners the collaboration was long-term and fruitful, whereas with others we failed to facilitate the relationship. We will focus on improving the analysis and conclusions of this.

Figure 3 visualizes the journey of the participating doctoral students, but it does not define the Tuckman's phases. However, it would be highly interesting to visualize how individuals reacted to these phases and the process of moving to the next phase. At this point, the figure focuses on visualizing the diversity among the students and the need for flexible supervision models.

2.12. What are the reasons why some students start the process higher on the scale than others? It is interesting that COVID affected students differently and this seems worthy of some discussion and explanation.

Response: These differences visualize how the students have different assumptions and preparedness for the process. Those two students with a beginning grade close to zero were 1) a female student who did her master thesis in the same research group, thus the change from student to an employed researcher was confusing, and 2) a foreign female who moved to the country to do the dissertation. Before their Ph.D. studies, she worked in the industry, and she also holds a non-engineering master's degree, which challenged her onboarding into the technical school. Those students whose starting grade was higher were male, holding a master's degree in engineering: two of them were employed by a ministry or a research institute, and two of them did their master's degree at the same unit.

We agree that it is highly interesting how different students respond to the pandemic differently – we will focus more on this in the revised paper.

2.13. Provision of supporting data for any statements made is essential. E.g. page 11, "Journey mapping results highlight how doctoral students enjoyed the most those papers where they were working in an interdisciplinary team".

Response: Good point – however, these observations were made in interviews and journey mapping. More supporting information will be provided. However, due to privacy and ethical issues, we cannot publish individual journey maps, but we will provide summaries and examples of those.

2.14. "At the beginning of their studies, the position of not being the corresponding author was found to increase their belongingness and psychological safety".

Response: More supporting information and insights behind these conclusions will be provided.

2.15. "Regarding the growth as a researcher, ownership of their work, and an established role in the community, the 3-to-12-month research visits were found valuable." From where have these findings been obtained and how many students do they relate to (e.g. interview with student No. X in 202X; workshop no X).

Response: More supporting information and insights behind these conclusions will be provided.

2.16. In section 3.5, some discussion of whether the authors feel a fixed duration project in which all researchers start and finish together would generate a stronger collaboration than on ongoing, rolling project in which research start and finish at different times would be interesting.

Response: Very important point! This project-approach and full funding was one of our key ideas for the project – we will focus on critical analysis on these impact on doctoral education.

2.17. Section 3.6 seems unnecessary and could be removed.

Response: We will consider removing this section.

**Additional comments**

2.18. In the current form of the manuscript, the interdisciplinary aspects are not prominent. Therefore, consider changing the title, to replace "interdisciplinary" with "collaborative".

Response: For us, interdisciplinary research is a) the motivation for collaboration and b) the outcome of our research. Collaboration is the tool and ways of working that strengthen and initiate interdisciplinary research and impact, and thus we will not make changes in the manuscript based on this comment – however, we will clarify this mindset.

2.19. Provide more details of the case study to give context e.g. where did funding come from and was collaborative research a specific objective of the funders, what was the background to the project, did the senior researchers have prior experience of working together etc.

Response: Good point – we will provide more information about the setting and also discuss what outcomes this exceptional project can deliver to all doctoral education communities.